

# Ideas and perspectives: New research examples of autumnal
# climate change ecology
**Ulf Büntgen[1,2,3], and Paul J. Krusic[1,4]**
[1]Department of Geography, University of Cambridge, CB23EN Cambridge, UK
[2]Swiss Federal Research Institute WSL, Zürcherstrasse 111, 8903 Birmensdorf, Switzerland
[3]CzechGlobe and Masaryk University, Kotlářská 2, 61137 Brno, Czech Republic
[4]Navarino Environmental Observatory, 24001 Messenia, Greece

*Correspondence to:* U. Büntgen (ulf.buentgen@geog.cam.ac.uk)




**Abstract.** Changes in autumnal climate affecting the diversity and productivity of the
ecosphere are arguably as important as vernal climatic changes. Motivated by a recent call for
more research on the biological and ecological consequences of autumnal climate change
(Gallinat et al., 2015), we present three examples of innovative biogeoscience, employing
novel datasets and methodologies, which refine our ability to monitor the physiological
functioning and ecosystem performance during autumn. Drawn from recent research in
wildlife biology (big-game hunting), wood anatomy (tree-ring formation) and mycology
(mushroom inventory), these studies provide original insights that contribute to an improved
understanding of how varying environmental and climatic conditions impact the phenology,
productivity and diversity of different organisms in autumn.





**1 Big-game hunting**
Warming-induced range shifts along altitudinal and latitudinal gradients have been reported
for many plant and animal species around the world (Parmesan and Yohe, 2003; Thomas et
al., 2004; Lenoir et al., 2008; Harsch et al., 2009; Chen et al., 2011; Gottfried et al., 2012;
Pauli et al., 2012). The mobility and behavioral plasticity of large animals, however,
complicates detection of climate-induced population movements. Long-term, massively
replicated and geographically detailed hunting records can supplement traditional animal
tracking studies (Kays et al., 2015). For instance, the Swiss canton of Grisons has amassed
>230,000 harvest locations of four ungulate species (Büntgen et al., 2017). Carefully collected
since 1991, this inventory contains year-to-year and decadal niche tracking of free-ranging
ibex, chamois, red deer and roe deer populations in higher elevations, late in the year. The
significant upward trend in ranging habits over the last two decades coincides with warmer,
snow-free and vegetation-rich, conditions in September and October. Such findings will help
improve awareness of the interconnectivity of the full annual cycle, including the return to
winter ranges (Rivrud et al., 2016).

**2 Tree-ring formation**
Though wood formation in many extra-tropical species occurs during most of the warm
season, several plant physiological processes occur at the end of the growing season. Their
precise assessment has greatly improved following recent advances in quantitative wood
anatomy (Steppe et al., 2015). State-of-the-art studies combining high-resolution dendrometer
measurements with micro-anatomy have found xylem lignification can persist through
autumn (Cuny et al., 2015). Thus, autumnal conditions can stimulate and prolong woody
biomass production, leaving a fingerprint on the intra-annual course of the global carbon
cycle (Piao et al., 2008). The application of wood anatomical studies, particularly in
environments with strong and regular summer droughts such as the Mediterranean, could help





identify moisture-controlled metabolic processes and ecophysiological reactions during the
formation of tree rings, thereby enabling the separation of different development stages from
anatomical traits.

**3 Mushroom inventories**
Rapid emergence, short lifespans, and non-photoperiodic constraints (Körner and Basler,
2010), make mushroom fruiting bodies ideal indicators of changes in late growing season
conditions. Inter-annual and multi-decadal variations in the abundance of autumnal
sporocarps (productivity), as well as the intra-annual timing of their occurrence (phenology),
and species abundance (diversity), are closely related to the multifaceted interplay of biotic
(mycelium and host interaction) and abiotic (environment and climate) factors (Boddy et al.,
2014). Experimental findings, local observations, national inventories and their continental-
scale compilations allow autumnal mushroom 'fruit body' dynamics to be reconstructed. Over
seven million sporocarp records, representing >10,000 fungal species from nine countries,
have been drawn from various scientific and citizen-science projects (Andrew et al., 2017). In
addition to providing evidence of warming-induced spatiotemporal shifts in autumn
mushroom phenology (Kauserud et al., 2012), this pan-European mycological inventory
offers unique macro-ecological opportunities to assess how fungal communities interact with
their environment. Exploring how fungal fruit body productivity and diversity is linked to
biotic and abiotic factors, including tree growth, as well as climate variation and nitrogen
deposition (Büntgen and Egli, 2014; Andrew et al., 2016; Van Strien et al., 2017),
respectively, will provide new biological and ecological insights during autumn.

A non-traditional resource that can provide important mushroom-related data for

autumnal climate change research, are governmental emergency services. Poison centers, such
as the Swiss National Poisons Information Centre delivers 24-hour/7-days-a-week nationwide
free medical advice. Since its establishment in 1966, the center has registered over one





million poison-related inquiries with around one percent of all cases attributed to mushrooms
(Schenk-Jäger et al., 2016). Comparison between these >12,000 mushroom-related calls with
survey information from the Swiss National Data Centre for Biodiversity demonstrates the
ability of poison center data to capture spatiotemporal patterns of fungal phenology,
productivity and diversity (Schenk-Jäger et al., 2016).

**4 What's next?**
By providing timely examples of research initiatives that further a better understanding of
biological and ecological responses to autumnal conditions (Gallinat et al., 2015), we hope to
encourage diversity and creativity in future studies. For instance, there is a multitude of
aquatic organisms that have life histories stored in distinct seasonal increments (Cole and
Fairbanks, 1990; Morrongiello, et al., 2012; Black et al., 2014; Reynolds, et al., 2016).
Complementary to terrestrial plant growth, information recorded in long-lived fish, bivalve
and coral species can reflect autumnal and even winter signals at high temporal resolution
(Black et al., 2017).

Moreover, we agree with Williams et al. (2015) about the biological and ecological

importance of winter climate change. Knowledge of the intensity and duration of climate
variability during winter is particularly critical for higher latitude and altitude ecosystems,
where the impacts of winter temperature and precipitation on snow cover persist through most
of the year. Although varying between organisms and habitats, cold season trends and
extremes may alter chilling requirements, frost injury, energy and water balance, phenology
and community interactions. At the same time, winter warming generally exceeds that during
other months, with implications not only on the annual temperature cycle (Duan et al., 2017)
and the Earth's carbon balance (Piao et al., 2008; Friend et al., 2014), but also by creating a
temporal mismatch between the biological requirements of different ecosystem components
and climate (Williams et al., 2015; Marra et al., 2016).



Future research on climate change ecology should consider the effects of changing
temperature and hydroclimate (precipitation and drought) in autumn and winter. Emphasis
should be given to investigations of the temporal synchronization of climate variability and
species-specific biological demands. Future efforts should also consider mining the whole
range of non-traditional, environmental inventories and metrics that exist today. The
application of (process-based) mechanistic models (Friend et al., 2014; Yang et al., 2017),
capable of detecting interactive influences and nonlinear factors affecting physiological
alternations, and diversity in organisms throughout the year, are still in their infancy.

*Author contributions.* U. Büntgen developed the idea and wrote the paper together with P.J.
Krusic.

*Acknowledgements.* Dr. Jeff Diez (University of Riverside, USA), Dr. Andrew Friend
(University of Cambridge), Dr. Amanda Gallinat (Boston University), Prof. Christian Körner
(University of Basel), and Dr. Andrew Liebhold (USDA Forest Service, USA), kindly
commented on earlier versions of this paper. U.B. received funding from the Ministry of
Education, Youth and Sports of CR within the National Sustainability Program I (NPU I; GN
LO1415), and additional support was provided by NSF grant 0909541. We are particularly
thankful to all colleagues that made their data available.

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
