# Peer review of "Ideas and perspectives: New research examples of autumnal"

_Biogeosciences, 2017_

## Referee Comment (RC1) · F. Babst (Referee) · 11 Aug 2017

This perspectives article highlights three innovative approaches to study seasonal ecosystem responses to climate change. The authors emphasize the need to better understand autumnal and hibernal processes that have so far remained less studied compared to vernal phenology. Seasonality in ecosystem processes is a timely topic and I agree that "diversity and creativity in future studies" (l. 87) are needed to address it.

General comments:

[Figure]

The manuscript reads well, but I have a hard time fully grasping its message. The call for studies on ecosystem processes in autumn and winter is in itself not new. By framing the manuscript around this topic, the authors mostly second earlier papers, e.g. by Gallinat or Williams (as cited), that have advertised the same issues in more detail. This is unfortunate, because the research examples provided in sections 1-3 are innovative and this novelty is somewhat lost in a known framework.

My main suggestion is to de-emphasize the "autumn-winter message" and instead emphasize the merits of creative research approaches to tackle the full seasonality in climate change impacts. The three examples (animal habitat range, xylogenesis, and mushroom phenology) are not limited to the autumn season, but can provide valuable information for other seasons as well. Such information can nicely complement widely established data streams, such as $CO_2$-flux measurements or remotely sensed observations of vegetation dynamics. The paper would then advocate (even stronger than it does already) for the integration of a series of complementary (and so far underused) data sources to really understand the phenology and dynamics of ecosystems and how they respond to climate change. The authors could provide a list of possible such resources (including the ones in sections 1-3 and others) that would be inspiring and useful for researchers.

Specific comments:

l. 39: The "full annual cycle" of what? Do you mean seasonal migration here?

l. 47: I suggest replacing "micro-anatomy" with "cell-level measurements"

l. 48: I suggest adding . . ."favorable" autumn conditions. . .

l. 54: A link to productivity could be drawn here via wood density. Favorable autumn conditions may result in denser wood also earlier in the annual ring (Franceschini et al. 2012, Holzforschung) and thus more biomass per volume.

L. 94-97: This statement deserves a reference.

L. 107-108: This sentence basically summarizes my main suggestion (above) and could be somehow reflected in the title of the paper.

L. 108-111: The authors have not talked about mechanistic modeling earlier in the text and this ending feels very detached from the rest of the manuscript. I suggest either removing this statement or then making modeling a more inherent part of the paper.

---

## Referee Comment (RC2) · A. S. Gallinat (Referee) · 11 Sep 2017

This paper highlights how three novel data sources have been used to research the effects of climate change on autumn and winter ecology. The authors discuss the use of (1) records from big-game hunting to measure shifting ranges that coincide with warmer autumn conditions, (2) high-resolution observations of wood anatomy to measure late-season woody biomass production, and (3) mycological inventories and data from poison centers to measure fungal productivity, diversity, and phenology. I agree with the authors that innovative methods such as these are important to the future study of ecological shifts in the autumn season.

General comments: While I appreciate the message that autumn and winter ecology should receive more focus in biogeoscience research, that does not seem to be the true goal or added value of this paper. The value this paper adds is to explore the feasibility and potential of novel, creative data sources for autumn research. To make that goal clear and to accomplish it, I believe this paper requires some reframing and additional information. I recommend the following changes:

1) The addition of an introduction would help to outline the specific problem/s this paper aims to solve. For instance– the effects of climate change on autumn and winter are relatively unknown, and traditional/historical data sets are limited. The authors suggest, therefore, that creative, novel data sources be used for autumn and winter ecology, and provide three examples of such data sources, including proof of their value to climate change research, future opportunities for their use, and unique features and biases of the data. This context would help to nest the three examples within the paper as just that– *examples* within the greater context of novel data sets, that might help the reader to identify additional creative data sources and identify the potential and biases of those data sets. Currently the examples come off as the main message of the paper, rather than as support for a larger, cohesive, original message.

2) Connecting the three examples with a common format would help to remind the reader of the main message they illustrate. For instance, if the message is that novel data sources have demonstrated value, future possibility, and unique quirks that should be understood for optimal use, I would suggest the following format for each example: I. a description of the data source II. a summary of what has been found so far using the data (including specific information– for instance, what is the magnitude/extent of the range shifts detected from big-game records?) III. a description of what is left to be discovered from these records, or similar records (what else could we learn from big-game records? What other game records could be used to fill gaps in our knowledge of autumn and winter ecology?) IV. what are the unique biases and limitations of the data (with all novel data sources, this is an important question for researchers to consider–

for instance, are calls to poison centers a robust record of mushroom phenology or simply of foraging phenology?).

3) If the goal is to encourage researchers to identify additional novel resources for studying autumn and winter ecology, the conclusion could be reframed (away from what we do not know about autumn and winter) to explicitly point researchers toward other novel data sources, and/or to suggest a list of questions researchers might ask to determine the potential and limitations of their own unconventional data sources.

What I've described above is one potential direction for this paper that would be both valuable and original. If the authors prefer to go in another direction, it will still be important to identify a clear message that ties the three examples together, and to be explicit and detailed about what has already been demonstrated using these data sources and what is left to glean from them.

Specific comments:

L30-31: Mobility and behavioral plasticity also *allow* us to detect climate-induced population movements. Perhaps stating what other factors populations might move in response to, or being more specific about how detection is complicated, would help to clarify this sentence. It is also not clear whether this is about migration or range shifts.

L37: What is the magnitude of the shift?

L38-40: This sentence is vague; what full annual cycle is this in reference to? How will these findings improve awareness, and for whom?

L44: Better to explicitly state the physiological processes here, otherwise the point is vague until two sentences later.

L47-48: How far into autumn can xylem lignification persist? Specifics will help to justify the value of this data source.

L66: What is the duration of these phenology records? Are they all wild observations?

L67-68: What magnitude of shift? For how many species? Any additional details here will, again, strengthen the evidence that this is a valuable data source.

---

## Author Comment (AC1) · 11 Oct 2017

Dear Editors

We are thankful to Armanda S. Gallinat and Flurin Babst for both their fair and constructive evaluation of our article. We carefully considered all comments and suggestions, and improved the manuscript accordingly. All minor and major changes are outlined in the detailed point-by-point response below. As a result, we believe that the revised work – herein already provided as an electronic supplement with all changes marked in yellow – is now suitable for publication, and will appeal to a wide, interdisciplinary audience.

Please do not hesitate contacting us if anything remains unclear and/or further information is needed. We appreciate you considering this Ideas and perspectives contribution in Biogeosciences and look forward to hearing from you soon.

On behalf of the authors; Yours respectfully - ulf buentgen

Point-by-point response letter

F. Babst (Referee; received and published, 11 Aug 2017) This perspective article highlights three innovative approaches to study seasonal ecosystem responses to climate change. The authors emphasize the need to better understand autumnal and hibernal processes that have so far remained less studied compared to vernal phenology. Seasonality in ecosystem processes is a timely topic and I agree that "diversity and creativity in future studies" (l. 87) are needed to address it.

Many thanks for this overall positive and encouraging evaluation of our work.

The manuscript reads well, but I have a hard time fully grasping its message. The call for studies on ecosystem processes in autumn and winter is in itself not new. By framing the manuscript around this topic, the authors mostly second earlier papers, e.g. by Gallinat or Williams (as cited), that have advertised the same issues in more detail. This is unfortunate, because the research examples provided in sections 1-3 are innovative and this novelty is somewhat lost in a known framework. My main suggestion is to de-emphasize the "autumn-winter message" and instead emphasize the merits of creative research approaches to tackle the full seasonality in climate change impacts. The three examples (animal habitat range, xylogenesis, and mushroom phenology) are not limited to the autumn season, but can provide valuable information for other seasons as well. Such information can nicely complement widely established data streams, such as $CO_2$-flux measurements or remotely sensed observations of vegetation dynamics. The paper would then advocate (even stronger than it does already) for the integration of a series of complementary (and so far, underused) data sources to really understand the phenology and dynamics of ecosystems and how they respond to climate change. The authors could provide a list of possible such resources (including the ones in sections 1-3 and others) that would be inspiring and useful for researchers.

We re-centred the manuscript around the merits of more innovative research approaches to address the full seasonal cycle in modern attempts of climate change biology/ecology. The thoroughly revised article, now including an additional 16 references, is not only longer but also more nuanced.

l. 39: The "full annual cycle" of what? Do you mean seasonal migration here?

This part of the manuscript has been complete re-written and expanded to "Such findings underscore the advantage of considering climate and its influence on environmental conditions throughout the year. By the same token, early-year census data – from which autumnal hunting quotas are derived – could be mined for resolving connections between population density, harvest intensity and climate variability. Thus, a more complete picture of the external drivers of wildlife performance, including inter-annual changes in species-specific returns to winter ranges (Rivrud et al., 2016) is obtained.".

l. 47: I suggest replacing "micro-anatomy" with "cell-level measurements"

Changed "State-of-the-art studies combining high-resolution dendrometer readings with cell-level measurements have found xylem lignification of conifer species in northeastern France to persist into late autumn/early winter (Cuny et al., 2015).".

l. 48: I suggest adding . . . "favorable" autumn conditions . . .

Done.

l. 54: A link to productivity could be drawn here via wood density. Favorable autumn conditions may result in denser wood also earlier in the annual ring (Franceschini et al. 2012, Holzforschung) and thus more biomass per volume.

Added "Consequently, our ability to connect short-term seasonal climate variations and weather extremes with intra-annual fluctuations in wood quality and quantity has dramatically increased (Battipaglia et al., 2016; De Micco et al., 2016).".

L. 94-97: This statement deserves a reference.

Added "(Williams et al., 2015)".

L. 107-108: This sentence basically summarizes my main suggestion (above) and could be somehow reflected in the title of the paper.

We changed the title to "Pursuing climate change ecology throughout the year". Moreover, we added a more general 'Introduction' paragraph and 16 new references.

L. 108-111: The authors have not talked about mechanistic modeling earlier in the text and this ending feels very detached from the rest of the manuscript. I suggest either removing this statement or then making modeling a more inherent part of the paper.

We re-moved and improved the sentence to a much earlier position (under 'Tree-ring'formation) "Following recent advances in quantitative wood anatomy (Steppe et al., 2015), and improvements in process-based plant physiological modeling (Yang et al., 2017), our understanding of the circumstances that control the precise timing of lignification has greatly improved.", and added a new concluding statement "Future efforts should also consider mining the whole range of non-traditional, environmental inventories and metrics that exist today, or even planned to be available in due course. Quantifying the effects of seasonal climate on those biological controls regulating yearly growth patterns can only improve the efficacy of (process-based) mechanistic models by providing valuable details of how seasonal-specific conditions and responses are inter-correlated throughout an organisms' life cycle.".

A. S. Gallinat (Referee; received and published, 11 Sep 2017) This paper highlights how three novel data sources have been used to research the effects of climate change on autumn and winter ecology. The authors discuss the use of (1) records from big-game hunting to measure shifting ranges that coincide with warmer autumn conditions, (2) high-resolution observations of wood anatomy to measure late-season woody biomass production, and (3) mycological inventories and data from poison centers to measure fungal productivity, diversity, and phenology. I agree with the authors that innovative methods such as these are important to the future study of ecological shifts in the autumn season.

We are pleased to read these lines.

While I appreciate the message that autumn and winter ecology should receive more focus in biogeoscience research, that does not seem to be the true goal or added value of this paper. The value this paper adds is to explore the feasibility and potential of novel, creative data sources for autumn research. To make that goal clear and to accomplish it, I believe this paper requires some reframing and additional information. I recommend the following changes: 1) The addition of an introduction would help to outline the specific problem/s this paper aims to solve. For instance– the effects of climate change on autumn and winter are relatively unknown, and traditional/historical data sets are limited. The authors suggest, therefore, that creative, novel data sources be used for autumn and winter ecology, and provide three examples of such data sources, including proof of their value to climate change research, future opportunities for their use, and unique features and biases of the data. This context would help to nest the three examples within the paper as just that– *examples* within the greater context of novel data sets, that might help the reader to identify additional creative data sources and identify the potential and biases of those data sets. Currently the examples come off as the main message of the paper, rather than as support for a larger, cohesive, original message.

We added a new introductory paragraph "1 Background and motivation", which now provides a wider context for our examples.

2) Connecting the three examples with a common format would help to remind the reader of the main message they illustrate. For instance, if the message is that novel data sources have demonstrated value, future possibility, and unique quirks that should be understood for optimal use, I would suggest the following format for each example: I. a description of the data source II. a summary of what has been found so far using the data (including specific information– for instance, what is the magnitude/extent of the range shifts detected from big-game records?) III. a description of what is left to be discovered from these records, or similar records (what else could we learn from big-game records? What other game records could be used to fill gaps in our knowledge of autumn and winter ecology?) IV. what are the unique biases and limitations of the data (with all novel data sources, this is an important question for researchers to consider– for instance, are calls to poison centers a robust record of mushroom phenology or simply of foraging phenology?).

Again, we re-structured the entire paper, including an introductory paragraph, and added 16 further citations/references.

3) If the goal is to encourage researchers to identify additional novel resources for studying autumn and winter ecology, the conclusion could be reframed (away from what we do not know about autumn and winter) to explicitly point researchers toward other novel data sources, and/or to suggest a list of questions researchers might ask to determine the potential and limitations of their own unconventional data sources. What I've described above is one potential direction for this paper that would be both valuable and original. If the authors prefer to go in another direction, it will still be important to identify a clear message that ties the three examples together, and to be explicit and detailed about what has already been demonstrated using these data sources and what is left to glean from them.

We hope that the revised – longer and more nuanced – manuscript version provides a much clearer message, and is thus suitable for publication.

L30-31: Mobility and behavioral plasticity also *allow* us to detect climate-induced population movements. Perhaps stating what other factors populations might move in response to, or being more specific about how detection is complicated, would help to clarify this sentence. It is also not clear whether this is about migration or range shifts.

Changed to "Although often complex at different spatiotemporal scales, the mobility and behavioral plasticity of large animals may offer an opportunity to detect climate-induced population movements throughout different parts of the year.".

L37: What is the magnitude of the shift?

We re-wrote and expanded this section "A species-specific upward trend in the ungulates' autumnal harvest locations between 1991 and 2003 coincides with a mean September-October temperature increase of 1.3 °C during the same period, which translates into more favorable, snow-free and vegetation-rich autumnal conditions. Linear regression slopes reveal statistically significant (p < 0.05) uphill shifts of 135, 95 and 79 m for ibex, chamois and red deer (Büntgen et al., 2017b), respectively. Such findings underscore the advantage of considering climate and its influence on environmental conditions throughout the year.".

L38-40: This sentence is vague; what full annual cycle is this in reference to? How will these findings improve awareness, and for whom?

This section has been completely re-written (see also previous response to referee 1).

L44: Better to explicitly state the physiological processes here, otherwise the point is vague until two sentences later.

Expanded "Though tree-ring formation in many extra-tropical species occurs during most of the warm season, several auxin-driven plant development processes (Vanneste and Friml, 2009), such as the thickening and lignification of xylem-cell walls, mainly occurs at the end of a growing season.".

L47-48: How far into autumn can xylem lignification persist? Specifics will help to justify the value of this data source.

Added "State-of-the-art studies combining high-resolution dendrometer readings with cell-level measurements have found xylem lignification of conifer species in north-eastern France to persist into late autumn/early winter (Cuny et al., 2015). The timing and duration of such processes strongly depends on the species, microenvironment, and climate.".

L66: What is the duration of these phenology records? Are they all wild observations?

Re-written "Despite mushrooms' smaller economic, social and ecological importance (Büntgen et al., 2017c), in comparison to plants and animals, over seven million in situ observations of wildlife mushroom fruiting bodies, representing >10,000 fungal species from nine countries spanning most of the 20th century (Andrew et al., 2017), have been drawn from various scientific and citizen-science projects.".

L67-68: What magnitude of shift? For how many species? Any additional details here will, again, strengthen the evidence that this is a valuable data source.

Added "In addition to providing evidence of warming-induced spatiotemporal shifts in autumn mushroom phenology – the mean annual day of fruiting has become several days to weeks later (Kauserud et al., 2012), a pan-European mycological inventory offers unique macro-ecological opportunities to assess how fungal communities interact with the environment (Büntgen and Egli, 2014), including symbiotic associations with their host vegetation (Büntgen et al., 2013).".

Please also note the supplement to this comment:
https://www.biogeosciences-discuss.net/bg-2017-265/bg-2017-265-AC1-supplement.pdf

―――――――――――――――――――

**Supplement:**

**Ideas and perspectives: Pursuing climate change ecology throughout the year**

**Ulf Büntgen[1,2,3], and Paul J. Krusic[1,4]**

[1]Department of Geography, University of Cambridge, CB23EN Cambridge, UK

[2]Swiss Federal Research Institute WSL, Zürcherstrasse 111, 8903 Birmensdorf, Switzerland

[3]CzechGlobe and Masaryk University, Kotlářská 2, 61137 Brno, Czech Republic

[4]Navarino Environmental Observatory, 24001 Messenia, Greece

*Correspondence to:* U. Büntgen (ulf.buentgen@geog.cam.ac.uk)

**Abstract.** Changes in autumnal climate affecting ecosphere diversity and productivity are arguably as important as winter, vernal and summer conditions. Motivated by the recent calls for more research on the biological and ecological consequences of autumnal, winter and full year climate change (Gallinat et al., 2015; Williams et al., 2015; Marra et al., 2016), we present three examples of innovative biogeoscience, employing novel datasets and methodologies, which refine our ability to monitor the physiological functioning and ecosystem performance during autumn. Drawn from recent research in wildlife biology (big-game hunting), wood anatomy (tree-ring formation) and mycology (mushroom inventory), these studies provide original insights that contribute to an improved understanding of how varying environmental and climatic conditions impact the phenology, productivity and diversity of different organisms in autumn.

**1 Background and motivation**

Many organisms are mainly active during the warm season. Our understanding of seasonal-specific biological and ecological responses to intra- and inter-annual environmental changes, including climate, is therefore biased. Novel data and methods from innovative biogeosciences, however, offer the possibility for extending climate change biology and ecology throughout the year. Large-scale, long-term surveys and crowdsourcing programs are a new and valuable source of seasonal information (Newman et al., 2012; Mills et al., 2015). When posing the right questions to the right persons, and applying the correct techniques and searching for allusive signals in hitherto unknown and putatively unsuitable archives (Isaac et al., 2014), citizen science projects can reveal novel and unexpected findings (Henderson et al., 2012).

Here, we present timely case studies from disparate disciplines that refine our ability to monitor ecosystem responses to seasonal-specific climate conditions. These examples from wildlife population ecology, wood anatomical-oriented dendroecology, and climate change mycology are intended to illustrate how innovative and interdisciplinary research on the phenology, productivity and diversity of organisms, during periods other than when it is most convenient, or when empirical evidence is most abundant, can resolve intra-annual processes affected by climate change. The recent maturity of massive datasets, from agency surveys to citizen science, offer an unprecedented opportunity for innovative experiments to extend climate change biology and ecology throughout the year.

**2 Animal migration**

Warming-induced range shifts along altitudinal and latitudinal gradients have been reported for many plant and animal species around the world (Parmesan and Yohe, 2003; Thomas et al., 2004; Lenoir et al., 2008; Harsch et al., 2009; Chen et al., 2011; Gottfried et al., 2012; Pauli et al., 2012). Although often complex at different spatiotemporal scales, the mobility and behavioral plasticity of large animals may offer an opportunity to detect climate-induced population movements throughout different parts of the year. For example, long-term, massively replicated and geographically detailed hunting records, can supplement traditional animal tracking studies (Kays et al., 2015). Since 1991, the Swiss canton of Grisons has amassed >230,000 harvest locations of four ungulate species (Büntgen et al., 2017b). This inventory reveals year-to-year and decadal niche tracking of free-ranging ibex, chamois, red deer and roe deer populations at higher elevations, late in the year. A species-specific upward trend in the ungulates' autumnal harvest locations between 1991 and 2003 coincides with a mean September-October temperature increase of 1.3 °C during the same period, which translates into more favorable, snow-free and vegetation-rich autumnal conditions. Linear regression slopes reveal statistically significant ($p < 0.05$) uphill shifts of 135, 95 and 79 m for ibex, chamois and red deer (Büntgen et al., 2017b), respectively. Such findings underscore the advantage of considering climate and its influence on environmental conditions throughout the year. By the same token, early-year census data – from which autumnal hunting quotas are derived – could be mined for resolving connections between population density, harvest intensity and climate variability. Thus, a more complete picture of the external drivers of wildlife performance, including inter-annual changes in species-specific returns to winter ranges (Rivrud et al., 2016) is obtained.

**3 Tree-ring formation**

Though tree-ring formation in many extra-tropical species occurs during most of the warm season, several auxin-driven plant development processes (Vanneste and Friml, 2009), such as the thickening and lignification of xylem-cell walls, mainly occurs at the end of a growing season. Following recent advances in quantitative wood anatomy (Steppe et al., 2015), and improvements in process-based plant physiological modeling (Yang et al., 2017), our understanding of the circumstances that control the precise timing of lignification has greatly improved. State-of-the-art studies combining high-resolution dendrometer readings with cell-level measurements have found xylem lignification of conifer species in north-eastern France to persist into late autumn/early winter (Cuny et al., 2015). The timing and duration of such processes strongly depends on the species, microenvironment, and climate. Favorable autumnal conditions can stimulate and prolong woody biomass production, leaving a fingerprint on the intra-annual course of the global carbon cycle (Piao et al., 2008). The application of wood anatomical studies, particularly in environments with strong and regular summer droughts such as the Mediterranean, could help identify moisture-controlled metabolic processes and ecophysiological reactions during the formation of tree rings, thereby enabling the separation of different development stages from anatomical traits. Consequently, our ability to connect short-term seasonal climate variations and weather extremes with intra-annual fluctuations in wood quality and quantity has dramatically increased (Battipaglia et al., 2016; De Micco et al., 2016).

**4 Mushroom production**

Rapid emergence, short lifespans, and non-photoperiodic constraints (Körner and Basler, 2010), make mushroom fruiting bodies ideal indicators of changes in late growing season conditions. Inter-annual and multi-decadal variations in the abundance of autumnal sporocarps (productivity), as well as the intra-annual timing of their occurrence (phenology), and species abundance (diversity), are closely related to the multifaceted interplay of biotic (mycelium and host interaction) and abiotic (environment and climate) factors (Boddy et al., 2014). Experimental findings, local observations, national inventories and their continental-scale compilations, allow seasonal- and species-specific mushroom 'fruit body' dynamics to be reconstructed. Despite mushrooms' smaller economic, social and ecological importance (Büntgen et al., 2017c), in comparison to plants and animals, over seven million *in situ* observations of wildlife mushroom fruiting bodies, representing >10,000 fungal species from nine countries spanning most of the 20th century (Andrew et al., 2017), have been drawn from various scientific and citizen-science projects. In addition to providing evidence of warming-induced spatiotemporal shifts in autumn mushroom phenology – the mean annual day of fruiting has become several days to weeks later (Kauserud et al., 2012), a pan-European mycological inventory offers unique macro-ecological opportunities to assess how fungal communities interact with the environment (Büntgen and Egli, 2014), including symbiotic associations with their host vegetation (Büntgen et al., 2013). Exploring how fungal fruit body productivity and species diversity is linked to biotic and abiotic factors, such as spore maturation and dispersion (Kauserud et al., 2011; Büntgen et al., 2017a), as well as climate variation and nitrogen deposition (Boody et al., 2014; Andrew et al., 2016; Van Strien et al., 2017), respectively, will provide new biological and ecological insights throughout the year.

Another non-traditional source of important mushroom-related data for seasonal climate change research, are governmental emergency services. Poison centers, such as the Swiss National Poisons Information Centre delivers 24-hour/7-days-a-week nationwide free medical advice (http://toxinfo.ch). Since its establishment in 1966, the center has registered over one million poison-related inquiries with around one percent of all cases attributed to mushrooms (Schenk-Jäger et al., 2016). Comparison between these >12,000 mushroom-related calls with survey information from the Swiss National Data Centre for Biodiversity (Senn-Irlet., 2010) demonstrates the ability of poison center data to capture spatiotemporal patterns of fungal phenology, productivity and diversity (Schenk-Jäger et al., 2016).

**5 What's next?**

By providing timely examples of research initiatives that further a better understanding of biological and ecological responses to autumnal conditions (Gallinat et al., 2015), we hope to encourage diversity and creativity in future studies. Such attempts, for instance, should consider the biological and ecological importance of all season, including winter climate change. Knowledge of the intensity and duration of climate variability during winter is particularly critical for higher latitude and altitude ecosystems (Williams et al., 2015), where the impacts of winter temperature and precipitation on snow cover persist through most of the year. Although varying between organisms and habitats, cold season trends and extremes may alter chilling requirements, frost injury, energy and water balance, phenology and community interactions. At the same time, winter warming generally exceeds that during other months, with implications not only on the annual temperature cycle (Duan et al., 2017) and the Earth's carbon balance (Piao et al., 2008; Friend et al., 2014), but also by creating a temporal mismatch between the biological requirements of different ecosystem components and climate (Williams et al., 2015; Marra et al., 2016).

In a similar vein, we cannot ignore the wide range of phenological indicators, such as the precise timing of bird migration (Jenni and Kéry, 2003), flower blossoming (Aono and Kazui, 2008), and wine harvest (Cook and Wolkovich, 2016), which have been used to obtain high-spatiotemporal-resolution data on biological and ecological responses to climatic and environmental trends and extremes throughout different seasons of the year. Moreover, aquatic organisms retain life histories in distinct seasonal increments (Cole and Fairbanks, 1990; Morrongiello, et al., 2012; Black et al., 2014; Reynolds, et al., 2016). For instance, the assessment of long-lived fish, bivalve and coral species can reveal autumnal and even winter signals at high temporal resolution (Black et al., 2017). Such data might be particularly valuable for supplementing insights from terrestrial archives to draw a more complete picture of biological and ecological responses throughout the year (Piermattei et al., 2017).

Curiosity-driven, proactive research on climate change ecology should consider the effects of changing temperature and hydroclimate (precipitation and drought) in all seasons. Emphasis should be given to investigations of the temporal synchronization of climate variability and species-specific biological demands. Future efforts should also consider mining the whole range of non-traditional, environmental inventories and metrics that exist today, or even planned to be available in due course. Quantifying the effects of seasonal climate on those biological controls regulating yearly growth patterns can only improve the efficacy of (process-based) mechanistic models by providing valuable details of how seasonal-specific conditions and responses are inter-correlated throughout an organisms' life cycle.

*Author contributions.* U. Büntgen conceived the idea and compiled the evidence. Both authors discussed the ideas and perspectives and jointly wrote the paper.

*Acknowledgements.* Dr. Flurin Babst (WSL, Switzerland), Dr. Jeff Diez (University of Riverside, USA), Dr. Andrew Friend (University of Cambridge), Dr. Amanda Gallinat (Boston University), Prof. Christian Körner (University of Basel), and Dr. Andrew Liebhold (USDA Forest Service, USA) kindly commented on earlier versions of this paper. U.B. received funding from the Ministry of Education, Youth and Sports of CR within the National Sustainability Program I (NPU I; GN LO1415), and additional support was provided by NSF grant 0909541. We are particularly grateful to our colleagues for making their data available.

**References**

Andrew, C., Heegaard, E., Halvorsen, R., Martinez-Pena, F., Egli, S., Kirk, P.M., Bässler, C., Büntgen, U., Aldea, J., Høiland, K., Boddy, L. and Kauserud, H.: Climate impacts on fungal community and trait dynamics. Fungal Ecol., 22, 17–25, 2016.

Andrew, C., Heegaard, E., Kirk, P., Bässler, C., Heilmann-Clausen, J., Krisai-Greilhuber, I., Kuyper, T., Senn, B., Büntgen, U., Diez, J., Egli, S., Gange, A., Halvorsen, R., Høiland, K., Nordén, J., Rustøen, F., Boddy, L. and Kauserud, H.: Big data integration: Pan-European fungal species observations' assembly for addressing contemporary questions in ecology and global change biology. Fungal Biol. Rev., 31, 88–98, 2017.

Aono, Y. and Kazui, K.: Phenological data series of cherry tree flowering in Kyoto, Japan, and its application to reconstruction of springtime temperatures since the 9th century. Int. J. Climatol., 28, 905–914, 2008.

Battipaglia, G., et al.: Structure and function of intra-annual density fluctuations: mind the gaps. Front. Plant. Sci., 7, 595, 2016.

Black, B.A., Sydeman, W.J., Frank, D.C., Griffin, D., Stahle, D.W., García-Reyes, M., Rykaczewski, R.R., Bograd, S.J. and Peterson, W.T.: Six centuries of variability and extremes in a coupled marine-terrestrial ecosystem. Science, 345, 1498–1502, 2014.

Black, B.A., Griffin, D., van der Sleen, P., Wanamaker, Jr. A.D., Speer, J.H., Frank, D.C., Stahle, D.W., Pederson, N., Copenhaever, C.A., Trouet, V., Griffin, S. and Gillanders, B.M.: The value of crossdating to retain high-frequency variability, climate signals, and extreme events in environmental proxies. Global Change Biol., 22, 2582–2595, 2016.

Boddy, L., Büntgen, U., Egli, S., Gange, A., Heegaard, E., Kirk, P., Mohammad, A. and Kauserud, H.: Climate variation effects on fungal fruiting. Fungal Ecol., 10, 20–33, 2014.

Büntgen, U. and Egli, S.: Breaking new ground at the interface of dendroecology and mycology. Trends Plant Sci., 19, 613–614, 2014.

Büntgen, U., Peter, M., Kauserud, H. and Egli, S.: Unraveling environmental drivers of a recent increase in Swiss fungi fruiting. Glob. Change Biol., 19, 2785–2794, 2013.

Büntgen, U., et al.: New insights into the complex relationship between weight and maturity of Burgundy truffles (*Tuber aestivum*). PLOS ONE, 12(1), e0170375, 2017a.

Büntgen, U., Greuter, L., Bollmann, K., Jenny, H., Liebhold, A., Galvan, J.D., Stenseth, N.C., Andrew, C. and Mysterud, A.: Elevational range shifts in four mountain ungulate species from the Swiss Alps. Ecosphere, 8(4), e01761. 10.1002/ecs2.1761, 2017b.

Büntgen, U., Latorre, J., Egli, S. and Martínez-Peña, F.: Socio-economic, scientific and political benefits of mycotourism. Ecosphere, 8(7), e01870. 10.1002/ecs2.1870, 2017c.

Chen, I.C., Hill, J.K., Ohlemüller, R., Roy, D.B. and Thomas, C.D.: Rapid range shifts of species associated with high levels of climate warming. Science, 333, 1024–1026, 2011.

Cole, J.E. and Fairbanks, R.G.: The southern oscillation recorded in the $\delta^{18}O$ of corals from

Tarawa atoll. Paleoceanograpy, 5, 669–683, 1990.

Cook, B.I. and Wolkovich, E.M.: Climate change decouples drought from early wine grape harvests in France. Nature Clim., 6, 715–720, 2016.

Cuny, H.E., Rathgeber, C.B.K., Frank, D.C., Fonti, P., Mäkinen, H., Prislan, P., Rossi, S.,

Martinez del Castillo, E., Campelo, F. et al.: Woody biomass production lags stem-girth increase by over one month in coniferous forests. Nature Plants, 1, 1–6, 2015.

De Micco, V., Campelo, F., De Luis, M., Bräuning, A., Grabner, M., Battipaglia, G. and

Cherubini, P. Intra-annual density fluctuations in tree rings: how, when, where, and why?

IAWA Journal, 37, 232–259, 2016.

Duan, J., Esper, J., Büntgen, U., Li, L., Xoplaki, E., Zhang, H., Wang, L., Fang, Y. and

Luterbacher, J.: Weakening of annual temperature cycle over the Tibetan Plateau since the 1870s. Nature Com., 8, 14008, 2017.

Friend, A.D., Lucht, W., Rademacher, T.T., Keribin, R., Betts, R., Cadule, P., Ciais, P., Clark,

D.B., Dankers, R., Falloon, P.D. et al.: Carbon residence time dominates uncertainty in terrestrial vegetation responses to future climate and atmospheric $CO2$. Proc. Natl. Acad.

Sci. USA, 111, 3280–3285, 2014.

Gallinat, A.S., Primack, R.B. and Wagner, D.L.: Autumn, the neglected season in climate change research. Trends Ecol. Evol., 30, 169–176, 2015.

Gottfried, M., et al.: Continent-wide response of mountain vegetation to climate change.

Nature Clim. Change, 2, 111–115. 2012.

Harsch, M.A., Hulme, P.E., McGlone, M.S. and R. P. Duncane, R.P.: Are treelines advancing? A global meta-analysis of treeline response to climate warming. Ecol. Lett.,

12, 1040–1049, 2009.

Henderson, S.: Citizen science comes of age. Frontiers Ecol. Environ., 10(6), 283–283, 2012.

Isaac, N.J., Strien, A.J., August, T.A., Zeeuw, M.P. and Roy, D.B.: Statistics for citizen science: extracting signals of change from noisy ecological data. Methods Ecol. Evo., 5,

1052–1060, 2014.

Jenni, L. and Kéry, M.: Timing of autumn bird migration underclimate change: advances in long-distance migrants, delays in short-distance migrants. Proc. R. Soc. Lond. B, 270,

1467–1471, 2003.

Kauserud, H., Heegaard, E., Halvorsen, R., Boddy, L., Høiland, K. and Stenset, N.C.:

Mushroom's spore size and time of fruiting are strongly related: is moisture important?

Biol. Lett., 7, 273–276, 2011.

Kauserud, H., Heegaard, E., Büntgen, U., Halvorsen, R., Egli, S., Boddy, L., Senn-Irlet, B.,

Greilhuber, I., Dämon, W., Sparks, T., Nordén, J., Høiland, K., Kirk, P., Semenov, M.

and Stenseth, N.C.: Warming-induced shift in European mushroom fruiting phenology.

Natl. Acad. Sci. USA, 109, 14488–14493, 2012.

Kays, R., Crofoot, M.C., Jetz, W. and Wikelski, M.: Terrestrial animal tracking as an eye on life and planet. Science, 343, 24781–24788, 2015.

Körner, C. and Basler, D.: Phenology under global warming. Science, 327, 1461–1462, 2010.

Lenoir, J., Gégout, J.C., Marquet, P.A., de Ruffray, P. and Brisse, H.: A significant upward shift in plant species optimum elevation during the 20th century. Science, 320, 1768–

1771, 2008.

Marra, P.P., Cohen, E.B., Loss, S.R., Rutter, J.E. and Tonra, C.M.: A call for full annual cycle research in animal ecology. Biol. Lett., 11, 20150552, 2016.

Mills, J.A., et al.: Archiving primarydata: solutions for long-termstudies. Trends Ecol. Evo.,

30(10), 581–589, 2015.

Morrongiello, J.R., Thresher, R.E. and Smith, D.C.: Aquatic biochronologies and climate change. Nature Clim. Change, 2, 849 – 857, 2012.

Newman, G., Wiggins, A., Crall, A., Graham, E., Newman, S. and Crowston, K.: The future
of citizen science: emerging technologies and shifting paradigms. Frontiers Ecol.
Environ., 10(6), 298–304, 2012.

Parmesan, C. and Yohe, G.: A globally coherent fingerprint of climate change impacts across
natural systems. Nature, 421, 37–42, 2003.

Pauli, H., et al.: Recent plant diversity changes on Europe's mountain summits. Science, 336,
353–355, 2012.

Piao, S., Ciais, P., Friedlingstein, P., Peylin, P., Reichstein, M., Luyssaert, S., Margolis, H.,
Fang, J., Barr, A. et al.: Net carbon dioxide losses of northern ecosystems in response to
autumn warming. Nature, 451, 49–52, 2008.

Piermattei, A., Urbinati, C., Tonelli, E., Eggertsson, Ó., Levanic, T., Kaczka, R., Andrew, C.,
Schöne, B.R. and Büntgen, U.: Potential and limitation of combining terrestrial and
marine proxy archives from Iceland. Glob. Planet. Change, 155, 213–224, 2017.

Reynolds, D.J., Scourse, J.D., Halloran, P.R., Nederbragt, A.J., Wanamaker, A.D., Butler,
P.G., Richardson, C.A., Heinemeier, J., Eirıksson, J., Knudsen, K.L. and Hall, I.R.:
Annually resolved North Atlantic marine climate over the last millennium. Nature Com.,
7, 13502, 2016.

Rivrud, I.M., Bischof, R., Meisingset, E.L., Zimmermann, B., Egil Loe, L. and Mysterud, A.:
Leave before it's too late: anthropogenic and environmental triggers of autumn migration
in a hunted ungulate population. Ecol., 97, 1058–1068, 2016.

Schenk-Jäger, K.M., Egli, S., Hanimann, D., Senn-Irlet, B., Kupferschmidt, H. and Büntgen,
U.: Introducing mushroom fruiting patterns from the Swiss National Poisons Information
Centre. PLOS ONE, 11(9), e0162314, 2016.

Senn-Irlet, B.: The use of a database for conservation–case studies with macrofungi. Mycol.
Balcanica, 7, 59–66, 2010.

Steppe, K., Sterck, F. and Deslauriers, A.: Diel growth dynamics in tree stems: linking anatomy and ecophysiology. Trends Plant Sci., 20, 335–343, 2015.

Thomas, C.D., et al.: Extinction risk from climate change. Nature, 427, 145–148, 2004.

Vanneste, S. and Friml, J.: Auxin: A trigger for change in plant development. Cell, 136 1005–

1016, 2009.

Van Strien, A.J., Boomsluiter, M., Noordeloos, M.E., Verweij, R.J.T. and Kuyper, T.W.:

Woodland ectomycorrhizal fungi benefit from large-scale reduction in nitrogen deposition in the Netherlands. J. Ecol., doi:10.1111/1365-2664.12944, 2017.

Williams, C.M., Henry, H.A.L. and Sinclair, B.J.: Cold truths: how winter drives responses of terrestrial organisms to climate change. Biol. Rev., 90, 214–235, 2015.

Yang, B., He, M., Shishov, V., Tychkov, I., Vaganov, E., Rossi, S., Ljungqvist, F.C.,

Bräuning, A. and Grießinger, J.: New perspective on spring vegetation phenology and global climate change based on Tibetan Plateau tree-ring data. Proc. Natl. Acad. Sci.

USA, doi/10.1073/pnas.1616608114, 2017.